# Drivers and trends of global soil microbial carbon over two decades

Guillaume Patoine [1,2✉], Nico Eisenhauer [1,2], Simone Cesarz [1,2], Helen R. P. Phillips [1,2,3,4,5], Xiaofeng Xu [6], Lihua Zhang[7] & Carlos A. Guerra [1,2]

Soil microorganisms are central to sustain soil functions and services, like carbon and nutrient cycling. Currently, we only have a limited understanding of the spatial-temporal dynamics of soil microorganisms, restricting our ability to assess long-term effects of climate and land-cover change on microbial roles in soil biogeochemistry. This study assesses the temporal trends in soil microbial biomass carbon and identifies the main drivers of biomass change regionally and globally to detect the areas sensitive to these environmental factors. Here, we combined a global soil microbial biomass carbon data set, random forest modelling, and environmental layers to predict spatial-temporal dynamics of microbial biomass carbon stocks from 1992 to 2013. Soil microbial biomass carbon stocks decreased globally by 3.4 ± 3.0% (mean ± 95% CI) between 1992 and 2013 for the predictable regions, equivalent to 149 Mt being lost over the period, or ~1‰ of soil C. Northern areas with high soil microbial carbon stocks experienced the strongest decrease, mostly driven by increasing temperatures. In contrast, land-cover change was a weaker global driver of change in microbial carbon, but had, in some cases, important regional effects.

[1] German Centre for Integrative Biodiversity Research (iDiv) Halle-Jena-Leipzig, Puschstraße 4, 04103 Leipzig, Germany. [2] Institute of Biology, Leipzig University, Puschstraße 4, 04103 Leipzig, Germany. [3] Department of Environmental Science, Saint Mary's University, Halifax, Nova Scotia, Canada. [4] Department of Life Sciences, Natural History Museum, London, UK. [5] Department of Terrestrial Ecology, Netherlands Institute of Ecology (NIOO-KNAW), 6700 AB Wageningen, Netherlands. [6] Biology Department, San Diego State University, San Diego, CA 92182, USA. [7] College of Life and Environmental Sciences, Minzu University of China, Beijing 100081, China. ✉email: guillaume.patoine@idiv.de

Soils account for the largest pool of terrestrial carbon[1] and provide essential functions and services to human and natural communities[2–4]. Soil microorganisms occupy a central role to sustain ecosystem functions by driving multiple biochemical processes, like decomposition, nutrient mineralization, nitrogen fixation, and carbon sequestration[5–8]. Microbial communities can be described in a number of ways—e.g., based on their abundance, diversity, and community composition—which are relevant facets of how these communities perform the resulting soil ecosystem functions[9]. Soil microbial biomass carbon (further as "microbial carbon") represents the amount of carbon in bacterial and fungal cells per unit of dry soil and is a valuable measure to quantify the size of the microbial community[10]. It can be estimated using a set of techniques developed since the 1970s, including fumigation, substrate-induced respiration, and phospholipid-derived fatty acids (PLFA)[11], and has been measured in a broad range of natural and managed ecosystems across the globe, offering good spatial and temporal coverage. Microbial carbon also acts as a critical carbon pool accounting for ~1% of soil organic carbon (but reaching much higher proportions in upper soil layers)[9], and a dynamic ecosystem component, having repercussions on climate feedbacks[12,13]. It is linked to soil functions like nutrient retention, enzyme activity, and aggregate stability[14–16], and has been used as a bioindicator of soil health and fertility in ecosystem assessments and modeling. Microbial carbon is therefore a key component of the microbial community that has been measured in numerous studies in many regions of the world, therefore providing an important knowledge base of the status of microbial communities globally.

To assure and support the continuation of soil ecosystem functions, we need to develop comprehensive assessments of long-term microbial carbon dynamics[17]. Although a loss in soil microbial carbon can undermine the provision of ecosystem functions[18], there is, as with other soil organisms, limited available data on microbial carbon changes over time[19–22]. Microbial communities are shaped by the interplay of geochemical and biological processes and are affected by specific environmental conditions, which makes it challenging to generalize observed patterns and develop a predictive understanding of their dynamics. Though, with sufficient observations originating from different regions and ecosystem types, it is possible to model microbial spatial-temporal dynamics and make informed extrapolations for large regions of the globe. Spatial-temporal models of soil biological communities can then help us to evaluate which areas of the globe may experience changes[23–25], and design appropriate management and conservation strategies[20].

At the global scale, soil microbial carbon depends primarily on geo-climatic factors and physicochemical soil characteristics[16,26]. Known parameters that affect soil microbial carbon include temperature, moisture, land cover, soil pH, and elevation[16,26]. Specifically, water availability and soil organic carbon content are crucial factors that promote microbial carbon and govern spatial patterns[16]. Soil pH also affects microorganisms in a non-linear fashion, where more neutral pH values lead to higher abundances and affects the community structure in terms of relative abundance of bacteria and fungi[27]. Of the known factors, climate and land cover are the most dynamic drivers of microbial carbon patterns, and have been heavily influenced by anthropogenic activity. The effects of climate and land-cover changes on soil communities, especially in interaction with each other, have not been sufficiently studied[3,7,28]. With an increase in environmental changes observed globally, soil communities and the resulting ecosystem functions are potentially at risk[18,29]. Particularly, the last decades were subject to important changes in land cover and vegetation types in many regions of the world[30], and were already visibly affected by climate change in terms of temperature and precipitation patterns[31]. Changes in both climatic conditions, as well as land cover, can have significant consequences for soil microbial carbon at the global and regional scales. The land-cover type defines a large part of the soil microbial community by changing the vegetation and carbon inputs[32]. In general, a higher land-use intensity leads to decreased soil microbial carbon[33], so more intensively managed soils with less vegetation and lower carbon content have less microbial carbon. Changes in land-cover type due to intensified usage (e.g., deforestation or changing grasslands to pastures) often affect the microbial community composition and cause a decrease in microbial carbon and diversity, often due to a reduction in above-ground plant biomass and soil carbon inputs[3,33,34].

To assess the sensitivity of microbial carbon to changes in climate and land cover, we evaluated temporal trends based on spatially-explicit predictions of soil microbial carbon globally. We used a global data set of microbial carbon[26] (Fig. 1) and global environmental layers to train a random forest model and generate global predictions of microbial carbon between 1992 and 2013. We then calculated rates of change in soil microbial carbon stocks globally and regionally, and tested which change in land cover or climate had the strongest effect on microbial carbon dynamics. We present yearly global maps of soil microbial carbon for that time period and the temporal trends for each predictable terrestrial location and aggregated by geographic region. While land-cover changes can result in rapid and dramatic effects on the local environment, they are globally less frequent than effects caused by climate change, which can affect larger regions simultaneously. We, therefore, expected climatic variables to have a stronger influence on soil microbial carbon dynamics due to their continuous effects on larger areas.

## Results and discussion

**Predictors of microbial carbon stocks**. We used a machine learning modeling approach to predict soil microbial carbon from a set of environmental covariates. To account for stochastic variability, we ran a set of models to assess the importance of environmental factors, which showed that the contribution of each variable to the model fit differed between runs, with some overlap between a number of them (Fig. 2b). Mean annual temperature was always the most important variable, with soil organic carbon and soil pH following. Clay content, precipitation, land-cover type, nitrogen content, and sand content contributed roughly equally to explaining variations in microbial carbon. Finally, NDVI and elevation had the lowest variable importance. Coniferous forests had the highest and most variable predicted values of microbial carbon (Supplementary Figs. 1, 2), which can be explained by high soil organic matter and a thick litter layer[26]. Tropical forests also had fairly high values of microbial carbon, while shrublands and croplands had the lowest values[26]. We used partial prediction response curves to evaluate the direction and range of effect of the predictor variables (Supplementary Figs. 1, 2). In agreement with the variable importance measure, variables that scored high often showed strong effects on the predicted microbial carbon values, while variables with a low variable importance score (e.g., elevation, NDVI, and sand content) only showed smaller responses. The only exception was for precipitation, which had a relatively high variable importance, although the response curves only showed a weak effect of precipitation for forests and grasslands, with limited effect on other land-cover types (Supplementary Fig. 2). The importance of precipitation might also indicate that this relationship involves interactions with other variables[7,28]. Overall, the differences in microbial carbon between land-cover types showed mostly similar patterns across the range of variables. Soil organic carbon

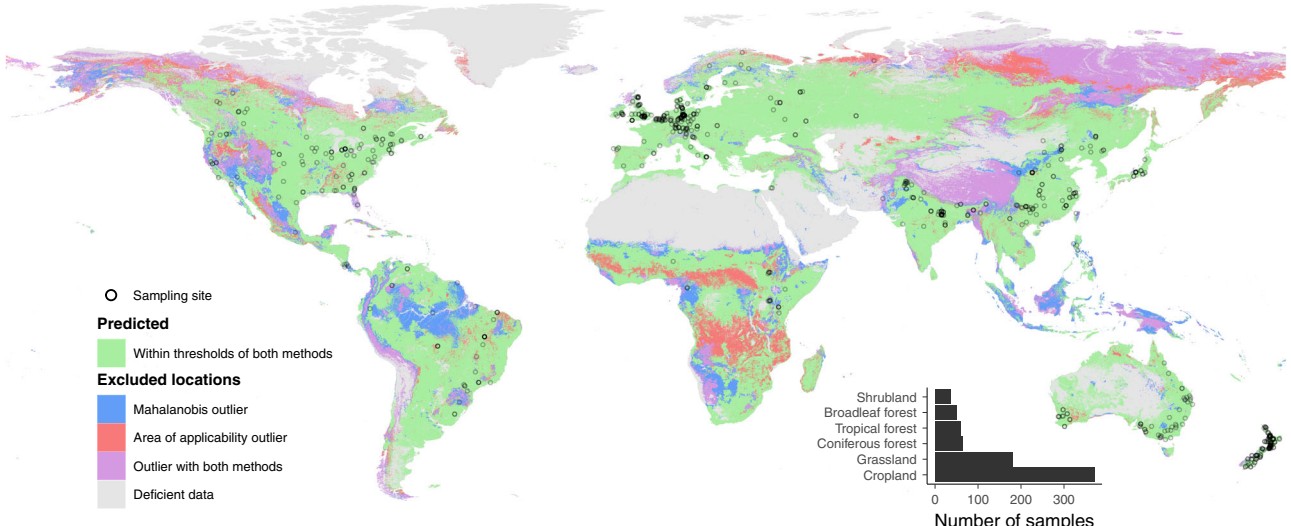

**Fig. 1 Sampling locations and environmental coverage analysis.** Sampling locations of the soil microbial carbon data set ($n = 762$). The extent of the environmental coverage of the global layers used (see Supplementary Table 3) as assessed by the Mahalanobis and Area of applicability methods with outliers represented in blue and red, respectively, and in purple for locations determined as outliers with both methods (see main text for details). Areas in gray are from regions with excluded land cover types (e.g., desert, glaciers). Further analyses were performed with the area in green that could be predicted with high confidence.

and nitrogen content had a positive and mostly linear effect on microbial carbon (Supplementary Fig. 1). In contrast, clay content, soil pH, and mean temperature had non-linear relationships, with high microbial carbon in the low range of these variables and a rapid decrease that reached an asymptote at low microbial carbon values for the higher portion of the range. Soil pH patterns showed a decrease in microbial carbon for values between 4.1 and 5.8, and a constant pattern between 5.8 and 8.6. Contrary to our expectations, we did not find a parabolic effect of soil pH on microbial carbon[26]. Instead, our model predicted higher values in very acidic soils with a pH below 5.2, which are rare globally and almost only found in central Amazonia. Similarly, locations with a clay content lower than 16.9% had higher values in microbial carbon, and then stabilized until 51.0%.

Mean temperature showed an interesting shift with much higher microbial carbon values with a mean annual temperature below zero, but had otherwise a limited effect on microbial carbon values in the rest of the range above zero up to 28.9 °C. Based on partial predictions (Supplementary Figs. 1–2), microbial carbon decreased monotonically with an increase in temperature (with all other variables fixed to their median), with the relationship being mostly stable for parts of the range. We observed an especially sharp decrease at around 0°C, which is in agreement with the patterns observed in the data. The reason for sites with a mean annual temperature below the freezing point to have higher microbial carbon stocks is not fully understood. This could be due to a regime shift in which microbial communities are in a semi-dormant state for a major part of the year[35]. Moreover, it could also be in part explained by the soil organic carbon content that follows a similar trend and accumulates in higher latitude soils[9], thus promoting higher microbial carbon stocks. Within these cold, high organic carbon soils, large microbial populations can be maintained, due to the low temperature that reduces metabolic requirements[35]. In contrast, at higher temperatures, metabolic activity increases and requires more resources and nutrients to maintain microorganisms alive. Experimental evidence is divided about the effects of warming on microbial carbon[18,36], highlighting the strong context-dependency of this relationship, although global observations show a clear pattern,

where low-temperature sites have higher soil microbial carbon stocks. Despite this uncertainty, there is a strong indication that a warming soil would tend to lose organic carbon[17,37], and subsequent patterns in microbial carbon can also be expected, because of the dependency on organic substrate[9,26,38]. These dynamics were observed in Melillo et al.[39], where the warming of sites in a mid-latitude forest ecosystem led to a decrease in soil carbon, followed by a decrease in microbial carbon[12].

Even with predictions being made for each grid location separately, microbial carbon values showed distinctive patterns and transitions over the globe (Fig. 2a). While temporal changes took place, broad spatial patterns were relatively constant over the range of years studied (Supplementary Movie 1). The highest microbial carbon stock values ranging from 1.50 to 7.00 t ha$^{-1}$ were found at high latitudes in the Northern Hemisphere in areas of coniferous forest. Tropical humid regions also showed high microbial carbon values between 0.50 and 1.50 t ha$^{-1}$ in the Amazon Rainforest and Central Africa. The main regions with low microbial carbon below 0.30 t ha$^{-1}$ were in Eastern South America, areas directly south of the Sahara Desert, East Africa, and most of Australia, all of which mostly correspond to shrublands. Cropland areas as seen in India were also predicted with low microbial carbon values ranging from 0.06 to 0.38 t ha$^{-1}$. A strong latitudinal gradient was visible for North America and Eurasia, with the highest microbial carbon stocks at high latitude, medium values in temperate ecosystems, and decreasing values towards the Equator. Positive coastal effects can also be observed, mostly on the Eastern South American and Australian coasts. In total, we estimated that there is 4.34 Gt of microbial carbon in the 5 to 15 cm layer for the predicted areas. Using the coefficient of variation calculated from the variability assessment set of models, we found that predictions made for the Amazon Basin, Northern Canada, and South-East Russia were more variable than for other regions (Supplementary Fig. 3a). Especially Western Europe, Central North America, and South-East Asia, however, showed high stability in the predictions between model runs.

**Drivers of change.** The analysis of the rate of change of microbial carbon stocks over time revealed that large regions of the globe experienced important changes in soil microbial carbon stocks

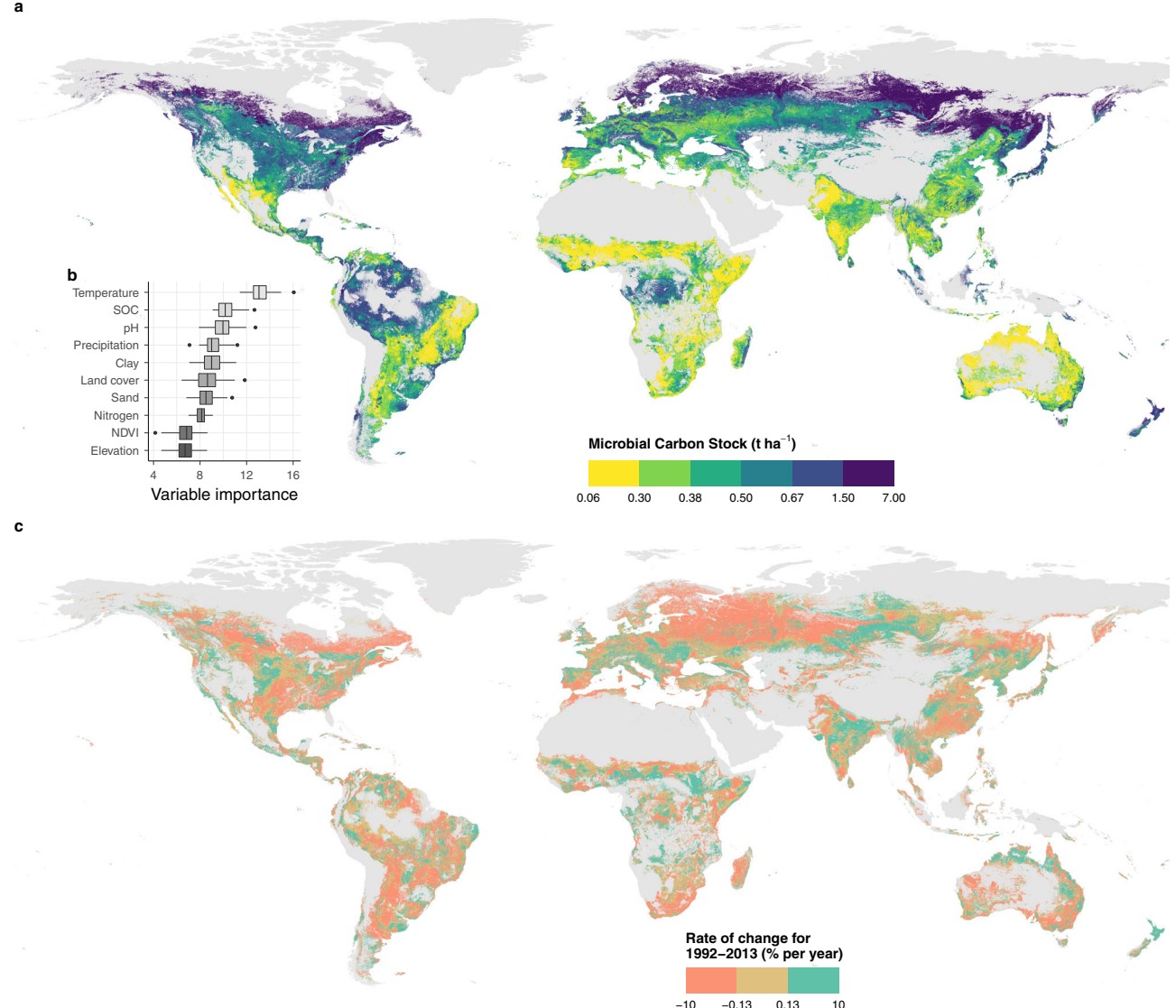

**Fig. 2 Microbial carbon stock spatial predictions and temporal trends. a** Microbial carbon stock predictions for 2013. **b** Variable importance from 100 random forest model runs, calculated by the mean decrease in accuracy after variable permutation. Variables were ordered by the median variable importance. SOC soil organic carbon, NDVI normalized difference vegetation index. Center line, median; box limits, upper and lower quartiles; whiskers, 1.5× interquartile range; points, outliers. **c** Relative microbial carbon stocks rate of change in percentage per year.

between 1992 and 2013, with contrasting patterns across areas, and overall larger regions showed a decrease rather than an increase in microbial carbon stocks (Fig. 2c and Supplementary Fig. 3b). To account for spatial differences in microbial carbon stocks, we calculated the relative rate of change in percentage for each location (Fig. 2c). When considering all predictable regions together, microbial carbon stocks in the 5–15 cm layer showed a decrease of 7.09 Mt per year, summing to 148.80 Mt between 1992 and 2013, or 3.4% of the global microbial carbon pool predicted (Supplementary Fig. 4a; $p = 0.038$). The main regions with a microbial carbon loss higher than 0.7 kg ha$^{-1}$ y$^{-1}$ were in Northern Canada and a large continuous region in North-Eastern Europe. These northern regions accounted for an important part of the global loss in microbial carbon stocks, with large areas that had both a high soil microbial carbon stock and a fast decrease (Figs. 3 and 4). Other areas of high loss were in the Amazon basin, Western Argentina, the USA East Coast, Southern South Africa, and South-East Russia. The main continuous region of microbial carbon increase above 0.7 kg ha$^{-1}$ y$^{-1}$ was in central

Russia, with smaller regions present in India, Europe, Central North America, and parts of Africa. Besides these general patterns, predictions vary at the local scale, and they consider the effects of parameters including soil properties, elevation, and land-cover type, which change between neighbor locations and affect the observed patterns. This is especially visible in the Americas, where both increases and decreases happen side-by-side.

Patterns in the relative rate of change have a lot in common with that of absolute change, with a few notable differences (Fig. 2c and Supplementary Fig. 3b). Both positive and negative stock changes in tropical and subtropical regions are more prominent in relative terms, as these regions typically have low microbial carbon stocks. Similarly, regions in Central Russia with high microbial carbon stocks show less decrease in relative terms. To assess how stable these trends are over time, we show the $p$ values of the rate of change for the 22 years (Supplementary Fig. 3c). The largest region with low $p$ values is associated with more significant trends in Western Russia, and corresponds to an

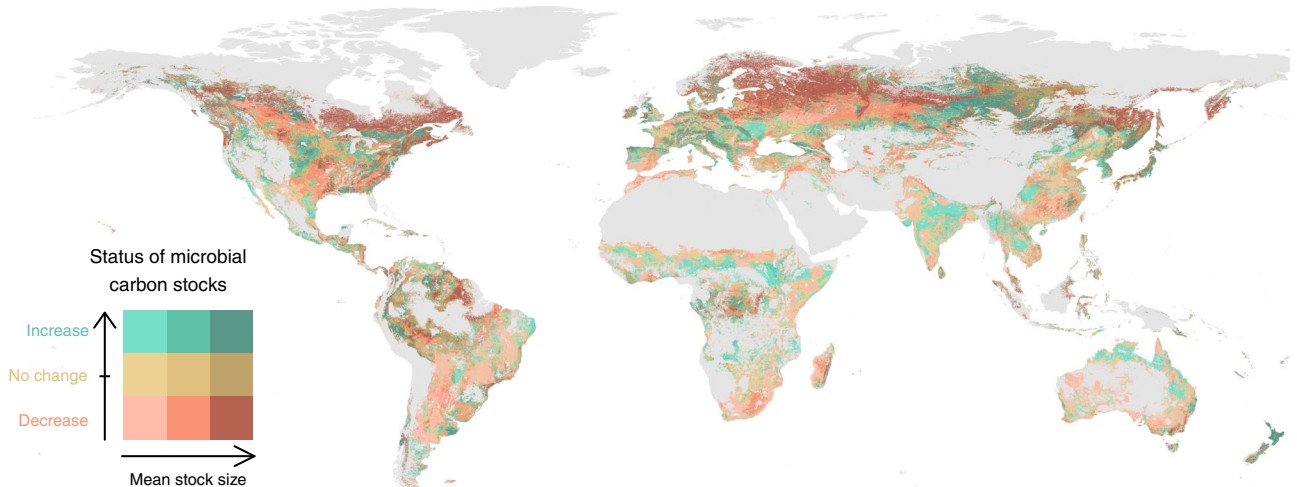

**Fig. 3 Status of microbial carbon stocks between 1992 and 2013.** Bivariate plot comparing the relative microbial carbon stock rate of change (% per year) with the amount of microbial carbon stock. The status groups were allocated using quantile distributions.

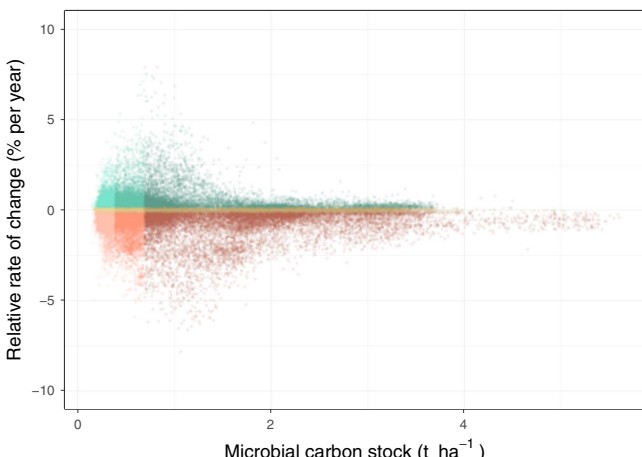

**Fig. 4 Distribution and classification of point values from the locations in Fig. 3.** The assignment of points into the 9 groups was performed using quantile distributions. Areas in dark red are especially vulnerable to climate and land-cover change.

area with a fast loss of microbial carbon. India and Central Russia show high $p$ values, and are informative of high variability compared to the strength of the signal. Considering that only up to 22 data points are available for each grid location and that especially climatic conditions vary considerably from year to year, $p$ values are only provided as a complementary assessment. We can summarize the global situation by combining the two maps of microbial carbon stocks and relative rate of change to categorize and define vulnerable locations that experienced a high loss of microbial carbon (Figs. 3 and 4), and where the provision of soil functions is potentially at risk.

It is informative to look at regional trends, by grouping grid locations using the Intergovernmental Science-Policy Platform on Biodiversity and Ecosystem Services (IPBES) sub-regions, and assessing regional-scale changes in microbial carbon stocks (Fig. 5, Supplementary Table 1). The main regions that contributed to microbial carbon loss were North America with a decrease of 62.49 Mt of microbial carbon and Eastern Europe with 60.88 Mt over the studied period, although both trends had high yearly variability and were non-significant. The region with the highest increase was North-East Asia with a gain of 4.49 Mt, but this change was also non-significant. The Caribbean was the only

region to show a significant increase in soil microbial carbon stocks over time (+2.1% over 22 y, $p = 0.017$), while significant decreases in stocks were found in North Africa (−4.1%, $p < 0.001$), South America (−1.7%, $p = 0.010$), Southern Africa (−2.6%, $p = 0.017$), and Central and Western Europe (−2.7%, $p = 0.034$; Supplementary Fig. 4a). Marginally significant decreases over the studied period were found in Western Asia (−2.5%, $p = 0.086$) and North America (−7.2%, $p = 0.093$). The 10 other regions showed no significant change in microbial carbon stocks, namely in Central Africa, Central Asia, East Africa and adjacent islands, Eastern Europe, Mesoamerica, North-East Asia, Oceania, South-East Asia, South Asia, and West Africa.

Climatic conditions and land-cover type are important aspects that affect soil microbial carbon stock dynamics. As we expected and has been found for other soil organisms (e.g., refs. [40–42]), climatic changes tended to have a stronger effect on microbial carbon stocks than those related to land cover (Fig. 5). Temperature patterns showed overall long-term warming in most regions, despite yearly variability, with a mean increase of 0.28 °C globally, promoting microbial carbon losses (Supplementary Figs. 5, 6). When looking at the separate effects of changes in climatic or land-cover variables (obtained by fixing the other set of variables), we found that globally, the decrease in microbial carbon was driven by climate change, with little effect on land-cover change (Fig. 5). Regionally, however, different resulting scenarios emerged from this analysis (Fig. 5 and Supplementary Fig. 4). The two groups of global change drivers had different influences on the predicted microbial carbon stocks depending on the region, with some being mostly affected by climate and others by land-cover change. The results of the interaction of both groups of variables either reinforced or masked the effect of each other. In many regions, one group of drivers was more prominent than the other and was mostly driving the general pattern. In a few cases, regions with non-significant effects in the full dynamic model showed significant effects of land-cover changes only (where climate variables are fixed), either with a positive effect (Central Asia and North America) or negative effect (South-East Asia). In addition, there were cases where the non-significant effects of climate and land-cover change combined to produce a significant overall effect (e.g., South America) and a special case in Western Asia (and less strongly in Oceania), where the significant effects of both driver types went in opposite directions, leading to overall non-significant effects. In these cases, the negative effect of one driver

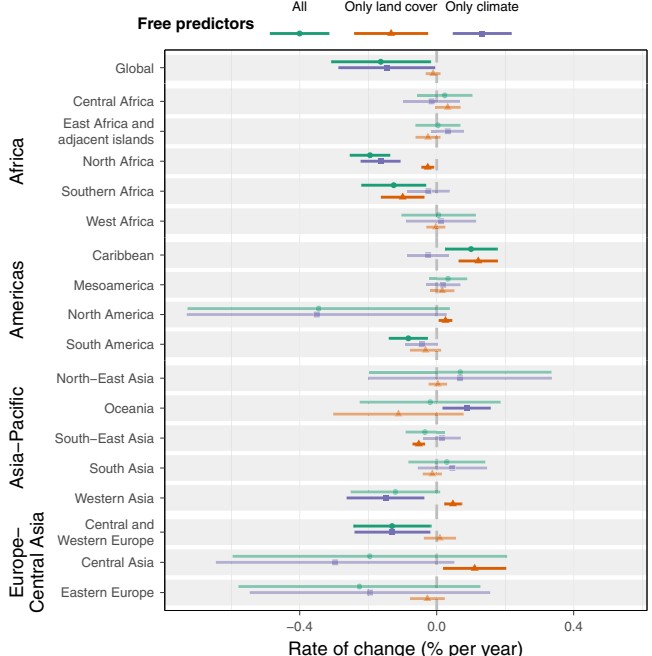

**Fig. 5 Regional and global drivers of trends in microbial carbon stocks.** Comparison of model predictions with either fixed climatic (temperature and precipitation) or land cover (land cover type and NDVI) variables. The central values are relative rates of change of soil microbial carbon stocks per year, calculated as the slope of the model fit, with 95% CI whiskers. Pale points are shown where the 95% CI crosses zero.

that could be predicted with high confidence, we performed an environmental coverage analysis based on two complementary methods that detect locations with environmental parameters (i.e., predictive variables) that are multi-dimensional outliers compared to the predictive data set. With the combined results of the two approaches to detect environmental outliers, we identified that the current knowledge of soil microbial carbon can be used to make predictions with confidence for locations representing 50.2% of terrestrial surfaces excluding glaciers (Fig. 1, Supplementary Fig. 7). Western Asia and North Africa were the regions with the lowest percentage area that could be predicted, with 5.1% and 10.6%, respectively (Supplementary Table 2). All other regions could be predicted for above 40% of their area, with Western and Central Europe having the largest proportion at 84.7%, followed by South America with 63.6% (Supplementary Table 2). As expected, highly sampled areas were included in regions that could be predicted with high confidence based on the environmental coverage assessment. Despite relatively low sampling density, South America, East Africa, and the western part of Eastern Europe were also regions with a large portion that could be predicted with confidence by both methods (Fig. 1, Supplementary Table 2). Outlier regions from both methods include North-East Russia and the Tibetan plateau, as well as a few smaller regions, mostly related to high latitude, elevation, or aridity. An important portion of the African continent consists of outlier regions, detected by both methods, but rarely in conjunction. The larger outlier areas in Africa mostly match regions of deciduous woodlands and savannah. A number of tropical rainforest regions were also excluded, including most of the Malay Archipelago, as well as central Amazonia.

The environmental coverage assessment highlights uncertainty in large regions, for which targeted sampling campaigns are needed to complement available data sets[21]. To further our understanding of microbial carbon dynamics across the globe, it is especially relevant to target regions with underrepresented environmental parameters, with high variability in microbial carbon, and that is expected to be most affected by changes in climate and land cover[45]. In addition, just as exploring under-sampled areas is important, repeated sampling at the same location provides valuable information for research, monitoring, and conservation efforts[46,47]. Like other global predictions based on statistical modeling[41,42,48], the results of this study represent general patterns and are informative to detect regional trends and should not be extrapolated to accurately estimate soil microbial carbon stocks at fine spatial and temporal scales, as local heterogeneity in soil properties and temporal climatic conditions can lead to variations in microbial carbon stocks.

is compensated by the positive effect of the other. Regions with a negative effect on at least one of the global drivers of change are especially vulnerable to being affected by the functions provided by the soil microbial community, especially in combination with high soil microbial carbon stocks[18]. These areas of vulnerability also often coincide with those most affected by environmental changes (Supplementary Figs. 5, 6). As global changes are expected to continue, and potentially accelerate[31], areas of vulnerability are likely to experience a continued decrease in soil microbial carbon stocks and a potential reduction or change of soil ecosystem functions. With our approach, we can look into region-specific drivers that led to modeled changes in microbial carbon stocks (Supplementary Figs. 3, 4). For example, Central and Western Europe experienced a decrease of 2.9% between 1992 and 2013, almost entirely driven by an increase in temperature of 0.64 °C. In this case, despite the increase in NDVI values over the period, land cover changes did not have much of an effect on microbial carbon dynamics. This region as a whole also experienced yearly fluctuations in precipitation, but showed no general trend over the studied period.

**Model evaluation and coverage.** The random forest model used for temporal predictions was validated by comparing the observed and predicted values of microbial carbon concentrations. The root-mean-square error (RMSE) was 65.0 mmol kg$^{-1}$, and the cross-validated $R^2$ for out-of-bag predictions was 0.40, while the overall $R^2$ was 0.90. The observed microbial carbon values correlated to the fitted values, with a Pearson's $r$ value of 0.59 (Fig. 6; $p < 0.001$). We believe that the clear definition of the geographical area of applicability of results is crucial for its proper interpretation, and that this type of assessment is often lacking from global predictions[43], especially as strong spatial biases in sampling locations are often observed[44]. To detect grid locations

**Ecological impacts and implications.** Soil microbial carbon is a crucial aspect of ecosystem health and services, and there are indicators that it is decreasing in many parts of the world. The effects of climate and land-cover change will continue to affect biological communities with potentially stronger effects in the years to come[49]. Targeted microbial communities that experience a decrease in microbial carbon can be affected in their ability to provide ecosystem functions, including food and material production, nutrient cycling, and carbon cycling. Microbial carbon represents a carbon pool that contributes to carbon sequestration and mediates carbon cycling[50,51]. As the amount of micro-organisms declines, these ecosystem functions are at risk to be affected negatively, and the continuity and magnitude of these services cannot be guaranteed[52]. With a decrease in soil microbial carbon, dark $CO_2$ fixation is also likely to decrease, therefore reducing the climate mitigation effects of soil microbial communities[53,54]. While some regions were more affected by

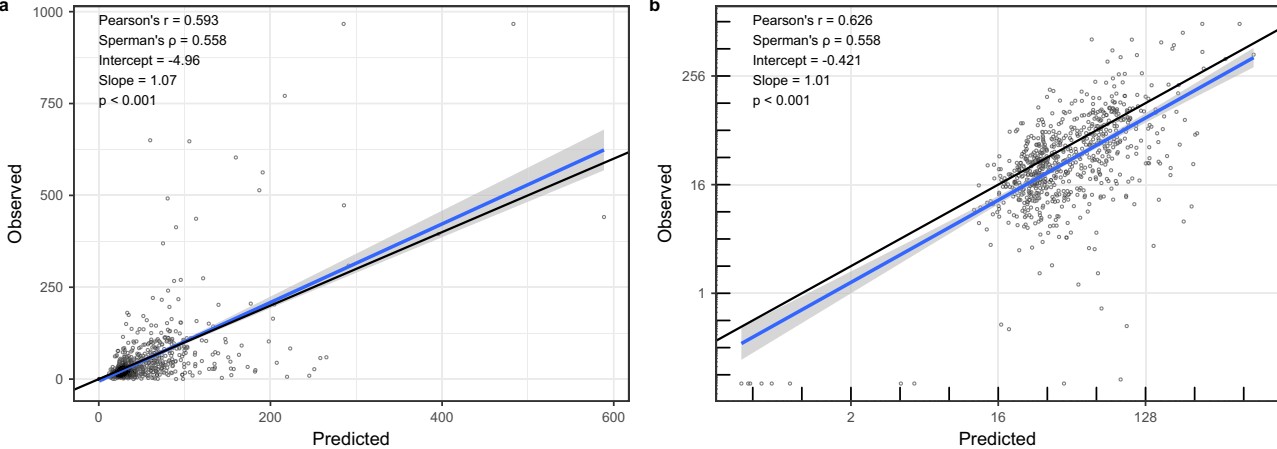

**Fig. 6 Microbial carbon model validation.** Relationship between observed and predicted values on a **a** linear scale and **b** log-scale. Reported statistical results from two-sided linear regressions. Gray areas represent 95% CI.

climatic changes, others were mostly driven by changes in land cover. In order to limit further losses in soil microbial carbon, both sets of drivers need to be addressed in cohesion, especially given the context-dependent effects of climate change[55]. Anthropogenic climate disruptions have led to regional changes in temperature and precipitation patterns that will continue to affect soil microbial communities[22]. While regulations and actions at the global scale are needed to slow anthropogenic climate change, local-scale management can address land degradation caused by changes in land cover that are detrimental to soil microorganisms and threaten soil functions[3,55]. Changes in land cover may take place naturally, e.g., as a response to climate changes, come from unregulated actions—as often seen in deforestation by small land-owners—or be the consequence of political decisions that affect land management at a larger scale. In that regard, land management can also be leveraged as a climate change mitigation and adaptation strategy, both to preserve microbial communities and sequester carbon[3,49]. In this stream, conservation, rewilding, and reforestation efforts focused on vulnerable areas can strongly contribute to supporting soil ecosystem functions and services, and soil communities should therefore be better integrated into conservation efforts[56].

While soil microbial communities continue to be studied, we can refine our mechanistic understanding of the belowground communities using diverse techniques that become increasingly accessible to describe additional aspects (e.g., diversity, community composition) and functionality, contributing to improving our understanding of this important ecosystem compartment and reduce uncertainty in global estimates[57–59], to complement microbial carbon measurements as the base measurement of microbial community size[9,21]. Currently, major global monitoring[46,60] as well as data mobilization and synthesis efforts[21] are taking place, that will help develop a more detailed perspective on the distribution, drivers, and trends of soil microbial communities and functions.

## Methods

### Data set description

*Microbial carbon.* The microbial carbon data set originated from a recent version of the global data set described in Xu et al.[26], updated with newly published data for 2012–2016. The data set contains microbial carbon values gathered from published literature with, when stated in the original paper, geographic coordinates, land cover description, sampling date, depth, and additional metadata. From that data set, we selected entries with soil microbial carbon measurements sampled at a mean depth above 30 cm. To allow the use of land-cover type as a predictor in further steps, we created a standardized land-cover type classification based on the ESA CCI data product[61] that harmonized classes across the microbial data set and the land-cover type gridded layers, based on the sampling sites' original description and geographic location from the papers

(see Supplementary Table 3). Sites from wetlands and bare areas (including both cold and warm deserts) were removed, as too few entries were available for proper statistical analysis and predictions (20 and 31 entries, respectively). The remaining six land cover categories used for further analysis were cropland, grassland, coniferous forest, tropical forest, broadleaf forest, and shrubland. Microbial carbon values measured from the same study were pooled together and averaged in cases where they were taken from the same location, year, and land-cover type, providing a total of 762 independent entries, with the number of sites per land cover ranging from 37 for shrublands to 370 for croplands (see inset in Fig. 1). This final data set contained 22 cases (57 entries) of time series, ranging between two and six years. Sampling locations spanned all continents except Antarctica with a higher concentration in Europe, North America, South-East Asia, Australia, and New Zealand, while lower representation in Western Asia, South America, and North Africa (Fig. 1).

*Global environmental layers.* We used gridded data sets representing climatic, environmental, geographic, land cover, vegetation, and soil property variables at the global scale to explain the observed patterns in microbial carbon (Supplementary Table 4). Soil property layers for organic carbon, total nitrogen, pH, sand proportion, and clay proportion were from the SoilGrids 250 m resolution data set[48]. The elevation layer was taken from WorldClim[62]. Global layers with yearly resolutions were used for climate, land cover, and vegetation cover. Yearly mean temperature and total precipitation were calculated from the monthly CHELSA time series[63,64]. The land-cover type layers from the ESA CCI data product[61] were reclassified to match the six categories from the microbial carbon data set (Supplementary Table 3). Land-cover types that could not be assigned to one of these categories were considered missing data (Fig. 1, Supplementary Table 3), which included bare areas, wetlands, and glaciers. To represent the amount of active vegetation, we compiled yearly estimates of the Normalized Difference Vegetation Index (NDVI) CDR acquired from NOAA's National Centers for Environmental Information[65,66]. As the NDVI daily data availability coverage was heterogeneous (i.e., clustered missing values), a monthly average was first taken, from which a yearly average was calculated, with an equal contribution from each month.

*Data extraction.* Values from all gridded environmental layers were extracted for each independent data point based on location, year, and sampling depth, where appropriate. The mean sampling depth was matched to one of the SoilGrids depth layers (0–5 cm, 5–15 cm, 15–30 cm), using the central layer of 5–15 cm when sampling depth was not specified. The sampling year was used to extract values from global data sets with a temporal component (i.e., climate and land cover, Supplementary Table 4). If the sampling year was not available from the paper, we used the five previous years before the publication year to extract the environmental values and average them. This range of years was chosen to match the pattern observed by other papers in the data set for the sampling-publishing year relationship. If the range of available years for a parameter did not cover the sampling year, we used the layer from the closest available year. Geographic coordinates for sampling locations were collected from the reviewed papers when available, or estimated based on the site name, research station, or the closest municipality. In order to standardize the extraction of values from global layers, correct for sampling locations that ended up inside water bodies or urban areas, and address the uncertainty in site geographic location, we selected the locations with matching land-cover types within 8 km of the provided sampling site and averaged the values using an inverse-distance weighting factor (i.e., the higher influence of closer locations).

**Environmental coverage**. To make spatial-temporal predictions of soil microbial carbon, we used the global layers of all model predictors for each year between 1992 and 2013, projected where needed, and resampled to a 0.05-degree grid resolution with the World Geodetic System 1984 as a coordinate reference system. To determine the spatial extent to which predictions could be made with high confidence, we used two complementary approaches that represent each grid location in multi-dimensional space, with one dimension per predictive variable, and compared it to the environmental coverage of the microbial carbon data set (Fig. 1). The first approach uses the Mahalanobis distance between the point given by all values of environmental variables of a grid location and the mean of the data set, once controlled for multicollinearity, which needs to be lower than an outlier threshold set at chisq = 0.975[20]. The second approach defines the Area of Applicability of the predictions, by comparing the dissimilarity index (DI; based on the distance to its closest neighbor in multi-dimensional space) of each grid location to the DI values from the training data[43]. Locations that have a DI higher than the threshold are considered outliers that cannot be predicted with high confidence. As they function under different principles, the two approaches complement each other well, and were therefore combined to define the spatial region where model predictions can be applied with confidence (Fig. 1). Locations are considered as environmental outliers from at least one of the methods were removed. Further analysis was performed using the remaining 2.6 million locations (Supplementary Table 2) on a 0.05 degree size grid with yearly predictions for the period 1992–2013.

**Modeling microbial carbon stock dynamics**. The effects of climate, land cover, vegetation, soil properties, and elevation on soil microbial carbon were studied with random forest modeling using R version 3.6.3[67], and the packages tidyverse[68], raster[69], caret[70], randomForest[71], and CAST[72] for data processing, model building, evaluation, and presenting results. Random forest is a modeling framework that accounts for non-linear effects and complex interactions between the predictors[73]. Model cross-validation was performed using 75% of the data set for training and the remaining part for validation at each resampling iterations. We used 500 trees for model training (ntree value) and 2 predictors sampled at each node for splitting (mtry value), chosen to minimize RMSE[73]. We trained one main "prediction model" to study the variable responses and to make yearly global predictions for the 22 years of this study. Predictions were made by extrapolating the relationships found in the model, using year-specific layers for all dynamic climatic and land cover layers. The model predictions could not be compared to an external microbial carbon data set, as no available additional data set could be found with sufficient spatial-temporal coverage that was not already included in the training data set. To evaluate the direction and shape of the response curve of microbial carbon for each variable separately, we looked at how model predictions changed over the range of values for each variable, while fixing the values of all other variables to the 0.25 and 0.75 quantiles. This approach provides a descriptive interpretation of the random forest model output that can be visualized and more easily interpreted. Due to the intrinsic stochasticity of random forest modeling, the measures of variable importance, as well as the predictions made from the trained models, can be sensitive to random changes in seed numbers and differ between runs[74]. To account for this variability, we also used a "variability assessment" set of 100 model runs specified with the same tuning parameters as the temporal prediction model to assess variable importance and calculate the coefficient of variation as the standard deviation of the microbial carbon predictions divided by the mean for each location.

The different methods used to measure microbial carbon are normally considered to be calibrated, so that they can be compared directly to each other[11]. To test this assumption and the potential effect of measurement methods on microbial carbon, we took two complementary approaches. We first reproduced the analysis using a reduced data set composed of only entries taken from fumigation methods, which was the most popular method, accounting for 72.7% of the entries. Using this reduced data set based on a unique measurement method, we reproduced the analysis workflow by training another random forest model and producing global predictions for the year 2013. The resulting predictions of the two data sets correlated with $R^2 = 0.97$. As a complementary analysis, we also ran a random forest model using the full data set, adding the measurement method as a model variable, and found that the measurement method was a poor predictor of microbial carbon and did not improve model fit substantially (RMSE = 66.2, cross-validated $R^2 = 0.41$; Supplementary Fig. 8). Taken together, the results of these sensitivity analyses indicate that there is no bias based on the method used.

We followed the approach described in Hengl et al.[75] to calculate microbial carbon stocks from the predicted concentrations considering the soil bulk density and fraction of coarse fragments. It has been reported that bulk density values might be overestimated, especially at high latitude[37]. We consider the risk of bias to be low for our predictions, considering that we excluded most locations at very high latitudes and limit our predictions to the 5–15 cm soil depth layer. We used all spatial grid locations with at least 10 years of predictions available and calculated the mean rate of change in microbial carbon over time using a separate linear regression model for each of the 2.6 million locations. To assess the global and regional patterns in soil microbial carbon stocks, we used the 17 IPBES sub-regions[76], combined all predicted grid locations, calculated a region-wise total amount of microbial carbon for each year, and studied the variation and trends over the 1992–2013 period. We used a yearly temporal resolution for the predictions without temporal correlation between predicted values and therefore cannot account for seasonal variations. We took this approach to describe microbial carbon dynamics as microbial communities can adapt fast enough in response to their environment so that legacy effects at the yearly scale are minimal. To test whether climate or land cover change had the strongest effect on microbial carbon changes, we compared the regional results with predictions made using fixed climatic and land cover variables over time. We, therefore, assigned the values from 1992 (the first predicted year) for either climatic (temperature and precipitation) or land cover (land-cover type and NDVI) variables.

**Reporting summary**. Further information on research design is available in the Nature Research Reporting Summary linked to this article.

## Data availability
The microbial carbon data set generated and used for analysis in this study has been deposited at https://zenodo.org/record/6645922 in the rawdata folder[77]. The global gridded data sets used in this study as covariates are publicly available (see Supplementary Table 4 for a full list).

## Code availability
The code used for analysis is available at https://zenodo.org/record/6645922 in the code folder[77].

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

## Acknowledgements

We gratefully acknowledge the support of the German Center for Integrative Biodiversity Research (iDiv) Halle-Jena-Leipzig funded by the German Research Foundation (FZT 118, 202548816). Helen RP Phillips has received funding from the European Union's Horizon 2020 research and innovation program under the Marie Skłodowska-Curie grant agreement No. 101033214 (GloSoilBio). The authors acknowledge support from the German Research Foundation (DFG) and Universität Leipzig within the program of Open Access Publishing.

## Author contributions

G.P., N.E., S.C., H.P., and C.G. designed the research. X.X. and L.Z. provided the data. G.P. performed the analysis and wrote the manuscript. N.E., S.C., H.P., and C.G.

contributed to the analysis and interpretation of the results. C.G. supervised the project. All authors contributed critically to the manuscript and gave final approval for publication.

## Funding

## Competing interests
The authors declare no competing interests.
