## [Peer Review File · Nature Communications]

Drivers and trends of global soil microbial carbon over two decadesREVIEWER COMMENTS

Reviewer #1 (Remarks to the Author):

I enjoyed reading this manuscript a lot. It is a compelling and novel assessment on how climate change and land cover change can affect changes in stocks of soil microbial biomass carbon over time and across space. The analysis and discussion of how the approach followed has limitations for some regions (due to lack of data) is a strength of this study, as it helps to interpret how the implications can differ depending of the region of the globe considered. I also have several comments and suggestions.

The major contribution of this study is the temporal assessment, as other previous studies have assessed the global distribution of soil MBC and its implications for soil C under climate change (Xu et al. 2013, Crowther et al 2020). First, authors addressed the effects of multiple environmental data on MBC (measured at each study in a particular year) with random forest modeling, then used these relationships to predict MBC over 1999-2013 using temporal datasets of environmental variables for this period. It is a robust approach, and authors cross-validated their results (using a subset of the data). However, an elegant supplement to this approach would be to gather MBC data from studies proving multiple years measurements, and relate those temporal empirical results with the predicted values.

The introduction is well framed and structured. Authors indicate microbial communities can be described using metrics of abundance, diversity and composition. Then they explain a couple of features that make MBC a good one for this study. I think they should better justify their metric of choice (MBC is the microbial metric with the largest spatiotemporal coverage in the literature, and its role in climate feedbacks has been assessed in current soil C models), and come back to this at the end of the first paragraph. On the other hand, MBC is a rather basic characterization of microbial communities, and soil C models and ecosystem services need to embrace more complex assessments of these soil communities (eg community composition, trait-based assessments) to reduce uncertainty and improve prediction. This overall context is important to avoid giving the wrong message of MBC as the ultimate metric, and it would be nice to deal with this in the discussion section. For instance, when comparing the importance of temperature (global importance) vs land cover (regional effects).

I have a concern on the methods, related with how authors dealt with the heterogeneity in the MBC dataset. For instance, authors say in the introduction that papers have used a suite of techniques (PLFA, SIR and fumigation). I wonder whether the method of choice in each paper has affected the influence of climate and land cover on MBC. I guess this analysis was also not performed in the original Xu et al paper, and would be nice to see it here. Authors seem to have dealt with other methodological covariates (eg only studies with soil depth 0-30 cm and mean values across years and sites), but not with MBC method.

It's always tricky to convert from MBC (or SOC%) to stocks, as soil bulk density is usually overestimated in global soil mapping products, rendering pretty high estimates particularly at high latitudes (García-Palacios et al 2021 Nature Revs Earth Environ). Although your study excludes many high latitude regions due to lack of MBC data, it would be nice to discuss this artefact in your MBC stocks estimations. For instance, your MBC stocks estimates are much higher than those of Xu et al, and I wonder whether the soil bulk density dataset used in each study drove such differences. L311-315 may be a good place to discuss these limitations, as the main goal of your study is not to predict fine-scale stocks but to compare global estimates across space and time.

Minor issues:

I suggest you use MBC as an acronym of microbial biomass carbon through the text instead of 'microbial carbon'

L36 six-words noun seems too much

L43 'predictable regions' may better give the message that some areas were not included in the analysis due to lack of data and/or model uncertainty

L45 can you quantify the overall climate warming for this 1992-2013 period? That would be helpful to understand the temperature effect on MBC.

L81 But soil MBC shows a clear latitudinal (associated with MAT) pattern.

L43 Can you link/convert this rate of MBC loss per year in overall SOC loss over this period so you can compare with the global soil C reservoir (ca. 1500 Gt C)?

L484-485 there is a typo in the journal name

Pablo García Palacios

Reviewer #2 (Remarks to the Author):

Soil microorganisms are very important in soil functions and services, under global climate change, it is critical to understand microorganisms' spatio-temporal variability and to evaluate the long-term effects of climate change and land-cover change on soil microorganisms. Based on a recent version of the global microbial carbon dataset, Patoine et al. used random forest modelling, with temperature, SOC, pH, precipitation, soil clay content, land-cover, soil sand content, nitrogen, NDVI, and elevation as predictors to estimate soil microbial carbon at global scale between 1992 and 2013. The authors further studied the trend of global soil microbial carbon and examined the drivers. The authors argue that soil microbial biomass carbon stocks decreased globally by 3.4% between 1992 and 2013, which is mainly related with global climate warming. The topic presented in this study is definitely of interest to the scientific community, and the main finding in this study is indeed noteworthy. The manuscript is well organized and generally easy to follow. However, I have one critical worry about the results in this study:

The main argument of this study is that the global soil microbial carbon is decreasing under global warming, however, from Supp. Figure 2, we can see that below $-1.8\text{ }^{\circ}\text{C}$, there is no relationship between soil microbial carbon and temperature, same for the data above $2.6\text{ }^{\circ}\text{C}$. The big difference of soil microbial carbon below -1.8 and above $2.6\text{ }^{\circ}\text{C}$ is interesting, why? In my opinion it should not due to temperature but other reasons. If not treat carefully, this may cause problem, for example, a region with mean temperature of about $-1.8\text{ }^{\circ}\text{C}$, under global warming, its mean temperature approaching $2.6\text{ }^{\circ}\text{C}$, then the model will predict significant decrease of soil microbial carbon, this is confirmed by Figure 3B that soil microbial carbon at northern high latitude area (mean temperature nearby $-1.8\text{ }^{\circ}\text{C}$) shows significant decrease pattern. The authors gave an example from a warming experiment to support their conclusion, however, results from one site is far from enough to support the global pattern. In addition, a meta-analysis (Figure 1a in Zhou et al. 2020, details of this reference please see below) showed that warming increased microbial alpha diversity, beta diversity, and microbial community structure, given those, it is unlikely soil microbial carbon will decrease under warming. Unfortunately, Zhou et al. 2020 does not compared soil microbial carbon under warming vs control. I suggest the authors contact with the corresponding author of Zhou et al. 2020 and ask whether their dataset has soil microbial carbon data. If this data is available, then comparing soil microbial carbon under warming and control and see whether soil microbial carbon decreasing under warming treatment. If the warming experiments also support soil microbial carbon is decreasing as temperature increasing, then it is a much strong evidence. However, if the opposite is the case, then the method in this study maybe pragmatic. Finally, what are the mechanisms to explain that soil microbial carbon is decreasing under global warming? For example, when a site's temperature increases from $-1.8\text{ }^{\circ}\text{C}$ to $2.6\text{ }^{\circ}\text{C}$, it should be a more friendly environment for the growth of microorganisms, why the microbial carbon will decrease? For those reasons, I am not convinced that soil microbial carbon is decreasing under global warming, and more evidences should be presented to support the argument.

Zhou, Zhenghu, Chuankuan Wang, and Yiqi Luo. "Meta-analysis of the impacts of global change factors on soil microbial diversity and functionality." *Nature communications* 11, no. 1 (2020): 1-10.

Some minor comments please see below:

Line 136-137 and Supp. Figure 2: why soil microbial carbon from coniferous forests are much higher than other vegetation types? From figure 1, the sample size is small for coniferous forests, are those samples reprehensive enough? And you used "Forest coniferous" in the figure legend, but in the main text, coniferous forests was used, please be consistent.

Line 154-156: An example from a site warming experiment is not enough to support the global pattern. A meta-analysis of warming vs control (Zhou et al. 2020) likely provides opposite results.

Please see my comments above.

Line 156-158: there are still not an agreement on whether soil carbon will decrease under global warming, please see those two papers below:

Crowther, Thomas W., Katherine EO Todd-Brown, Clara W. Rowe, William R. Wieder, Joanna C. Carey, Megan B. Machmuller, B. L. Snoek et al. "Quantifying global soil carbon losses in response to warming." *Nature* 540, no. 7631 (2016): 104-108.

Van Gestel, Natasja, Zheng Shi, Kees Jan Van Groenigen, Craig W. Osenberg, Louise C. Andresen, Jeffrey S. Dukes, Mark J. Hovenden et al. "Predicting soil carbon loss with warming." *Nature* 554, no. 7693 (2018): E4-E5.

Line 226: IPBES needs to be defined when it was first used.

Line 278: A r-squared of 0.40 between predicted values and measured values suggest that the performance of the random forest model is not very good!

Line 290: "outlier" appeared very abruptly, it should be briefly explained before you used it, otherwise the readers do not know what you are talking about.

Line 298: "two approaches to detect environmental outliers" should also be briefly introduced, otherwise, the readers have to move to the method section to figure it out.

Line 321-322: this is conflict with the results from (Zhou et al. 2020).

Line 557: some information of reference 24 are missing.

Line 708: what are short bars in the x and y axis, data density? However, from the data points, those bars do not like density.

Figure 3b: in the figure legend, land cover was used, but land-cover was used in other places, please check and be consistent in the whole manuscript.

Figure 5: what new information can the bivariate plot provide compared with figure 3c? In my opinion, this figure is not necessary, and a bivariate plot makes it harder to interoperate. Some other minor comments on this figure (may not important but want to point it out), 1) panel A does not have a figure frame but panel B has; 2) figure label "A" located outside of the frame (if have one) but panel "B" located inside the frame; and 3) why panel B is smaller than panel A?

Supp. Figure 2: here showed the results based on the model predictions, how about the results from the raw data (i.e., 762 samples shown in Figure 1)?

Could you please show the partial dependence plots for each of the 10 predictor variables? An example of the partial dependence plots please see the reference below:

Warner, D. L., Benjamin Bond-Lamberty, Jinshi Jian, Emma Stell, and Rodrigo Vargas. "Spatial predictions and associated uncertainty of annual soil respiration at the global scale." *Global Biogeochemical Cycles* 33, no. 12 (2019): 1733-1745.

REVIEWER COMMENTS

Reviewer #1 (Remarks to the Author):

I enjoyed reading this manuscript a lot. It is a compelling and novel assessment on how climate change and land cover change can affect changes in stocks of soil microbial biomass carbon over time and across space. The analysis and discussion of how the approach followed has limitations for some regions (due to lack of data) is a strength of this study, as it helps to interpret how the implications can differ depending on the region of the globe considered. I also have several comments and suggestions.

The major contribution of this study is the temporal assessment, as other previous studies have assessed the global distribution of soil MBC and its implications for soil C under climate change (Xu et al. 2013, Crowther et al 2020). First, authors addressed the effects of multiple environmental data on MBC (measured at each study in a particular year) with random forest modeling, then used these relationships to predict MBC over 1999-2013 using temporal datasets of environmental variables for this period. It is a robust approach, and authors cross-validated their results (using a subset of the data). However, an elegant supplement to this approach would be to gather MBC data from studies providing multiple years measurements, and relate those temporal empirical results with the predicted values.

Response: We thank the reviewer for their valuable comments. We also appreciate the suggestion to compare the predicted microbial carbon stock values with field measurements. First, it is to be noted that a few of the studies included in the microbial carbon dataset contain entries over multiple years. These are already considered in the model training and validation, which strengthens the analysis. We added in the text: "This final dataset contains 22 cases (57 entries) of time series, ranging between two and six years." Lines 400-401 .

The comparison to an external temporal microbial carbon dataset with sufficient data points would add a layer of validation, but we are not aware of an available dataset with sufficient spatial and temporal coverage that is not already included in the main microbial carbon dataset used. We added this information to the Methods section of the manuscript: "The model predictions could not be compared to an external microbial carbon dataset, as no available additional dataset could be found with sufficient spatial-temporal coverage that was not already included in the training dataset." Lines 478-481.

The introduction is well framed and structured. Authors indicate microbial communities can be described using metrics of abundance, diversity and composition. Then they explain a couple of features that make MBC a good one for this study. I think they should better justify their metric of choice (MBC is the microbial metric with the largest spatiotemporal coverage in the literature, and its role in climate feedbacks has been assessed in current soil C models), and come back to this at the end of the first paragraph. On the other hand, MBC is

a rather basic characterization of microbial communities, and soil C models and ecosystem services need to embrace more complex assessments of these soil communities (eg community composition, trait-based assessments) to reduce uncertainty and improve prediction. This overall context is important to avoid giving the wrong message of MBC as the ultimate metric, and it would be nice to deal with this in the discussion section. For instance, when comparing the importance of temperature (global importance) vs land cover (regional effects).

Response: We agree with the reviewer's comment in that other metrics are complementary and would be a critical next step. We did not want to discredit other measurements of microbial communities and modified the text in the introduction and discussion to put this in context: "Microbial carbon is therefore a key component of the microbial community that has been measured in numerous studies in many regions of the world, therefore providing an important knowledge base with good spatial coverage." Lines 69-71.

Moreover, we now point towards exciting current data synthesis initiatives that will help to take these mentioned next steps highlighted by the reviewer: "While soil microbial communities continue to be studied, we can refine our mechanistic understanding of the belowground communities using diverse techniques that become more available to describe additional aspects (e.g. diversity, community composition) and functionality, contributing to improve our understanding of this important ecosystem compartment and reduce uncertainty in global estimates⁵⁵⁻⁵⁷, to complement microbial carbon measurements as the base measurement of microbial community size^{9,21}. Currently, major global monitoring^{46,58} as well as data mobilization and synthesis efforts²¹ are being launched that will help develop a more detailed perspective on the distribution, drivers, and trends of soil microbial communities and functioning." Lines 370-378.

I have a concern on the methods, related with how authors dealt with the heterogeneity in the MBC dataset. For instance, authors say in the introduction that papers have used a suite of techniques (PLFA, SIR and fumigation). I wonder whether the method of choice in each paper has affected the influence of climate and land cover on MBC. I guess this analysis was also not performed in the original Xu et al paper, and would be nice to see it here. Authors seem to have dealt with other methodological covariates (eg only studies with soil depth 0-30 cm and mean values across years and sites), but not with MBC method.

Response: We thank the reviewer for their constructive comment. We performed additional analyses to address that point, and added a paragraph in the text and Supplementary Figure 2: “The different methods used to measure microbial carbon are normally considered to be calibrated, so that they can be compared directly to each other⁴¹. To test this assumption and the potential effect of measurement methods on microbial carbon, we took two complementary approaches. We first reproduced the analysis using a reduced dataset composed of only entries taken from fumigation methods, which was the most popular method, accounting for 72.7% of the entries. Using this reduced dataset based on a unique measurement method, we reproduced the analysis workflow by training another random forest model and producing global predictions for the year 2013. The resulting predictions of the two datasets correlated with $R^2 = 0.97$. As complementary analysis, we also ran a random forest model using the full dataset, adding the measurement method as model variable and found that the measurement method was a poor predictor of microbial carbon and did not improve model fit substantially (RMSE = 66.2, cross-validated $R^2 = 0.41$; Supplementary Fig. 2). Taken together, the results of these sensitivity analyses indicate that there is no bias based on the method used.” Lines 493-506.

It’s always tricky to convert from MBC (or SOC%) to stocks, as soil bulk density is usually overestimated in global soil mapping products, rendering pretty high estimates particularly at high latitudes (García-Palacios et al 2021 Nature Revs Earth Environ). Although your study excludes many high latitude regions due to lack of MBC data, it would be nice to discuss this artefact in your MBC stocks estimations. For instance, your MBC stocks estimates are much higher than those of Xu et al, and I wonder whether the soil bulk density dataset used in each study drove such differences. L311-315 may be a good place to discuss these limitations, as the main goal of your study is not to predict fine-scale stocks but to compare global estimates across space and time.

Response: We are grateful for the remark about bulk density biases and looked more into it to understand the issue, which is only mentioned briefly in the cited reference. We found information related to this for the SoilGrids layers in Hengl et al. (2017). To our understanding, this bias was, however, accounted for in the study, resulting in unbiased bulk density estimates, as shown in figure 8. In addition, bulk density estimate biases are more likely to occur at high latitude and for lower soil depths. As the predictable areas of our study generally exclude high latitude regions and are limited to the 5-15 cm depth, we consider that there is limited risk of including a strong effect of bulk density estimate biases. To acknowledge the potential issue, we added to the text: “It has been reported that bulk density values might be overestimated, especially at high latitude³⁷. We consider the risk of bias to be low for our predictions, considering that we excluded most locations at very high latitude and limit our predictions to the 5-15 cm soil depth layer.” Lines 510-513.

Xu et al. (2013) did not consider soil bulk density at the time of their analysis. Our microbial carbon concentration estimates (needed for stock calculation, but not presented in the paper) are comparable to those found by Xu et al. (2013).

Minor issues:

I suggest you use MBC as an acronym of microbial biomass carbon through the text instead of 'microbial carbon'

Response: We understand the reviewer's point of view. We considered their suggestion, but would like to minimize the use of acronyms, in order to improve the readability of the article.

L36 six-words noun seems too much

Response: We tried to simplify the text: "However, spatial-temporal dynamics of soil microorganisms are poorly understood, which limits our ability to assess long-term effects of climate and land-cover change on microbial roles in soil biogeochemistry. This study assesses the temporal trends in soil microbial carbon biomass and identifies the main drivers of biomass change regionally and globally to detect the areas sensitive to environmental factors." Lines 35-39.

L43 'predictable regions' may better give the message that some areas were not included in the analysis due to lack of data and/or model uncertainty

Response: We agree and changed the sentence: "Soil microbial biomass carbon stocks decreased globally by $3.4 \pm 3.0\%$ (mean \pm 95% CI) between 1992 and 2013 for the predictable regions [...]". Lines 42-44.

L45 can you quantify the overall climate warming for this 1992-2013 period? That would be helpful to understand the temperature effect on MBC.

Response: We now provide an estimate of global mean temperature change over the studied period to complement Supp. Figures 7 and 8 in the discussion: "Temperature patterns showed overall long-term warming in most regions, despite yearly variability, with a mean increase of 0.28°C globally, promoting microbial carbon losses (Supplementary Fig. 7-8)." Lines 265-267.

L81 But soil MBC shows a clear latitudinal (associated with MAT) pattern.

Response: We agree with the reviewer that this was a misleading statement and removed it: "Specifically, water availability and soil organic carbon content are crucial factors that promote microbial carbon and govern spatial patterns¹⁶." Lines 88-90.

L43 Can you link/convert this rate of MBC loss per year in overall SOC loss over this period so you can compare with the global soil C reservoir (ca. 1500 Gt C)?

Response: We appreciate this comment. We added the comparison in the text: "Soil microbial biomass carbon stocks decreased globally by $3.4 \pm 3.0\%$ (mean \pm 95% CI) between 1992 and 2013 for the predictable regions, equivalent to 149 Mt being lost over the period, equivalent to $\sim 1\%$ of soil C." Lines 42-44.

L484-485 there is a typo in the journal name

Response: We thank the reviewer for catching that and corrected the typo: "Upon acceptance of this manuscript, the microbial carbon dataset used for analysis will be deposited on a public data repository (e.g. Dryad), following the data accessibility guidelines of Nature Communications." Lines 531-533.

Pablo García Palacios

Reviewer #2 (Remarks to the Author):

Soil microorganisms are very important in soil functions and services, under global climate change, it is critical to understand microorganisms' spatio-temporal variability and to evaluate the long-term effects of climate change and land-cover change on soil microorganisms. Based on a recent version of the global microbial carbon dataset, Patoine et al. used random forest modelling, with temperature, SOC, pH, precipitation, soil clay content, land-cover, soil sand content, nitrogen, NDVI, and elevation as predictors to estimate soil microbial carbon at global scale between 1992 and 2013. The authors further studied the trend of global soil microbial carbon and examined the drivers. The authors argue that soil microbial biomass carbon stocks decreased globally by 3.4% between 1992 and 2013, which is mainly related with global climate warming. The topic presented in this study is definitely of interest to the scientific community, and the main finding in this study is indeed noteworthy. The manuscript is well organized and generally easy to follow. However, I have one critical worry about the results in this study:

The main argument of this study is that the global soil microbial carbon is decreasing under global warming, however, from Supp. Figure 2, we can see that below -1.8°C , there is no relationship between soil microbial carbon and temperature, same for the data above 2.6°C . The big difference of soil microbial carbon below -1.8 and above 2.6°C is interesting, why? In my opinion it should not due to temperature but other reasons. If not treat carefully, this may cause problem, for example, a region with mean temperature of about -1.8°C , under global warming, its mean temperature approaching 2.6°C , then the model will predict significant decrease of soil microbial carbon, this is confirmed by Figure 3B that soil microbial carbon at northern high latitude area (mean temperature nearby -1.8°C) shows significant decrease pattern. The authors gave an example from a warming experiment to support their conclusion, however, results from one site is far from enough to support the global pattern. In addition, a meta-analysis (Figure 1a in Zhou et al. 2020, details of this reference please see below) showed that warming increased microbial alpha diversity, beta diversity, and microbial community structure, given those, it is unlikely soil microbial carbon will decrease under warming. Unfortunately, Zhou et al. 2020 does not compared soil microbial carbon under warming vs control. I suggest the authors contact with the corresponding author of Zhou et al. 2020 and ask whether their dataset has soil microbial carbon data. If this data is available, then comparing soil microbial carbon under warming and control and see whether soil microbial carbon decreasing under warming treatment. If the warming experiments also support soil microbial carbon is decreasing as temperature increasing, then it is a much strong evidence. However, if the opposite is the case, then the method in this study maybe pragmatic. Finally, what are the mechanisms to explain that soil microbial carbon is decreasing under global warming? For example, when a site's temperature increases from -1.8°C to 2.6°C , it should be a more friendly environment for the growth of microorganisms, why the microbial carbon will decrease? For those reasons, I am not convinced that soil microbial carbon is decreasing under global warming, and more evidences should be presented to support the argument.

Zhou, Zhenghu, Chuankuan Wang, and Yiqi Luo. "Meta-analysis of the impacts of global change factors on soil microbial diversity and functionality." *Nature communications* 11, no. 1 (2020): 1-10.

Response:

We thank the reviewer for their constructive comments. As the reviewer noticed, the relationship between microbial carbon and temperature is non-linear. Overall, microbial carbon decreases monotonically with an increase of temperature, with the relationship being mostly stable for parts of the range. There is especially a sharp decrease around 0°C in the partial prediction plots, which is in agreement with the patterns observed in the data (see Supplementary Figure 4).

Zhou et al. (2020) provides valuable information related to our analysis. Figure 1 in Zhou et al. (2020) shows a positive effect of warming for beta diversity and community structure, but non-significant effects for alpha diversity (both richness and Shannon). In addition, Zhou et al. (2020) analysed the effect of warming on microbial carbon, shown on Figure 5, and found no significant effect of warming on the response ratio of microbial biomass. The dataset used by Zhou et al. (2020) was available with the paper and contained 42 entries with the response ratio of microbial biomass from soils that underwent warming treatment. We reanalysed these data and also found no significant effect of warming magnitude or duration on the response ratio of microbial carbon (both $p > 0.05$; see figures below). Another meta-analysis from Romero-Olivares et al. (2017) also found no significant effect of warming on microbial carbon. Many single studies found significant effects in either direction, which highlights the strong context-dependency of the warming effects on microbial carbon. While experimental evidence is divided about the effects of warming on microbial carbon, global observations show a pattern where low temperature sites often have higher microbial carbon stocks.

To clarify this point, we added in the text: “Based on partial predictions (Supplementary Fig. 3-4), microbial carbon decreased monotonically with an increase in temperature (with all other variables fixed to their median), with the relationship being mostly stable for parts of the range. We observed an especially sharp decrease at around 0°C, which is in agreement with the patterns observed in the data. The reason for sites with a mean annual temperature below the freezing point to have higher microbial carbon stocks is not fully understood. This could be due to a regime shift in which microbial communities are in a semi-dormant state for a major part of the year³⁵. Moreover, it could also be in part explained by the soil organic carbon content that follows a similar trend and accumulates in higher latitude soils⁹, thus promoting higher microbial carbon stocks. Within these cold, high organic carbon soils, large microbial populations can be maintained, due to the low temperature that reduces metabolic requirements³⁵. In contrast, at higher temperatures, the metabolic activity increases and requires more resources and nutrients to maintain microorganisms. Experimental evidence is divided about the effects of warming on microbial carbon^{18,36}, highlighting the strong context-dependency of this relationship, although global observations show a strong pattern, where low temperature sites have higher soil microbial carbon stocks. Despite this uncertainty, there is strong indication that a warming soil would tend to lose organic carbon^{17,37}, and subsequent patterns in microbial carbon can also be expected, because of the strong dependency on organic substrate^{9,26,38}. These dynamics were observed in Melillo et al.³⁹, where the warming of sites in a mid-latitude forest ecosystem led to a decrease in soil carbon, followed by a decrease in microbial carbon¹².” Lines 160-180.

Figure. Re-analysis of the dataset from Zhou et al. (2020), showing the effect of duration of warming (in years) and warming intensity on the response ratio of microbial biomass.

Some minor comments please see below:

Line 136-137 and Supp. Figure 2: why soil microbial carbon from coniferous forests are much higher than other vegetation types? From figure 1, the sample size is small for coniferous forests, are those samples representative enough? And you used "Forest coniferous" in the figure legend, but in the main text, coniferous forests was used, please be consistent.

Response: Indeed, coniferous forests have higher microbial carbon stocks than other land-cover types. This is also what is observed in the dataset, as described in Xu et al. (2013), considering that our “Coniferous forest” category encompasses both “Temperate coniferous forests” and “Boreal boreal” from the classification used by Xu et al. (2013), with the latter having especially high values. There were 65 data points for coniferous forest, well distributed geographically. The environmental coverage analysis that we performed assures that the predicted areas are well represented by the original dataset. To clarify that our predictions match the findings of Xu et al. (2013), we added the reference in the text : “Coniferous forests had the highest and most variable predicted values of microbial carbon, which can be explained by high soil organic matter and a thick litter layer²⁶. Tropical forests also had fairly high values of microbial carbon, while shrublands and croplands had the lowest values²⁶.” Lines 133-136.

In addition, we changed the forest labels as suggested in Figure 1, Supp. Figure 2, and Supp. Table 1 to “Coniferous forest”, “Broadleaf forest”, and “Tropical forest” to match with the main text.

Line 154-156: An example from a site warming experiment is not enough to support the global pattern. A meta-analysis of warming vs control (Zhou et al. 2020) likely provides opposite results. Please see my comments above.

Response: We agree with the point raised. We now provide probable mechanisms to support our findings. Please see the major comment above for text changes.

Line 156-158: there are still not an agreement on whether soil carbon will decrease under global warming, please see those two papers below:

Crowther, Thomas W., Katherine EO Todd-Brown, Clara W. Rowe, William R. Wieder, Joanna C. Carey, Megan B. Machmuller, B. L. Snoek et al. "Quantifying global soil carbon losses in response to warming." *Nature* 540, no. 7631 (2016): 104-108.

Van Gestel, Natasja, Zheng Shi, Kees Jan Van Groenigen, Craig W. Osenberg, Louise C. Andresen, Jeffrey S. Dukes, Mark J. Hovenden et al. "Predicting soil carbon loss with warming." *Nature* 554, no. 7693 (2018): E4-E5.

Response: We agree with the reviewer that there is no consensus about the effect of warming on soil carbon. We modified the text to clarify that the statement is not our opinion, but comes from the Melillo paper, where they observed the decrease in soil carbon: “These dynamics were observed in Melillo et al.³⁹, where the warming of sites in a mid-latitude forest ecosystem led to a decrease in soil carbon, followed by a decrease in microbial carbon¹².” Lines 178-180.

Line 226: IPBES needs to be defined when it was first used.

Response: Agreed. The acronym is now defined: “It is informative to look at regional trends, by grouping grid locations by Intergovernmental Science-Policy Platform on Biodiversity and Ecosystem Services (IPBES) sub-regions, and assessing regional scale changes in microbial carbon stocks.” Lines 245-247.

Line 278: A r-squared of 0.40 between predicted values and measured values suggest that the performance of the random forest model is not very good!

Response: We consider an r-squared value of 0.40 for the global model of soil microbial carbon concentration to be satisfactory, especially considering a root-mean-square error of $65.0 \text{ mmol} \cdot \text{Kg}^{-1}$, which is reasonable compared to the values ranging between 0.1 and almost 1000 $\text{mmol} \cdot \text{Kg}^{-1}$ (see Figure 2). As a comparison, van den Hoogen et al. (2019) had a similar r-squared for their random forest model: “mean cross-validation $R^2 = 0.43$, overall $R^2 = 0.86$ “. We now differentiate between the mean cross-validation r-squared and the overall r-squared, which is much higher: “The random forest model used for temporal predictions had a root-mean-square error (RMSE) of 65.0 and a cross-validated R^2 of 0.40 (overall $R^2 = 0.90$)”. Lines 301-302.

Line 290: “outlier” appeared very abruptly, it should be briefly explained before you used it, otherwise the readers do not know what you are talking about.

Response: We agree with the reviewer and added additional information to introduce the concept: “To detect grid locations that could be predicted with high confidence, we performed an environmental coverage analysis based on two complementary methods that detect locations with environmental parameters (i.e. predictive variables) that are multi-dimensional outliers compared to the predictive dataset. With the combined results of the two approaches to detect environmental outliers, we identified that the current knowledge of soil microbial carbon can be used to make predictions with confidence for locations representing 50.2% of terrestrial surfaces excluding glaciers (Figure 1, Supplementary Fig. 1).” Lines 307-314.

Line 298: “two approaches to detect environmental outliers” should also be briefly introduced, otherwise, the readers have to move to the method section to figure it out.

Response: Agreed. See previous response.

Line 321-322: this is conflict with the results from (Zhou et al. 2020).

Response: We are not sure that we understood the reviewer's comment, as the sentence on lines 321-322 is fairly general. In an attempt to address the comment, we reformulated: "Targeted microbial communities that experience a decrease in microbial carbon can be affected in their ability to provide ecosystem functions, including food and material production, nutrient cycling, and carbon cycling." Lines 347-350.

Line 557: some information of reference 24 are missing.

Response: We thank the reviewer for catching that and made the correction: "Xu, X., Thornton, P. E. & Post, W. M. A global analysis of soil microbial biomass carbon, nitrogen and phosphorus in terrestrial ecosystems. *Glob. Ecol. Biogeogr.* (2013)." Lines 610-611.

Line 708: what are short bars in the x and y axis, data density? However, from the data points, those bars do not like density.

Response: We agree that the log tick-marks in Figure 2 were not clearly described, nor very helpful, and we therefore removed them.

Figure 3b: in the figure legend, land cover was used, but land-cover was used in other places, please check and be consistent in the whole manuscript.

Response: We thank the reviewer for their remark. We tried to follow grammar rules as described in the Merriam-Webster, in which compound nouns are written separately, while compound adjectives require a hyphen. This is why we write, for example: "Land cover and climate affect soil systems." and "We used six land-cover categories." We corrected a few typos in the text for consistency.

Figure 5: what new information can the bivariate plot provide compared with figure 3c? In my opinion, this figure is not necessary, and a bivariate plot makes it harder to interpret. Some other minor comments on this figure (may not important but want to point it out), 1) panel A does not have a figure frame but panel B has; 2) figure label "A" located outside of the frame (if have one) but panel "B" located inside the frame; and 3) why panel B is smaller than panel A?

Response: We appreciate the reviewer's comment. Figure 3 was created to highlight the contrasting patterns in total microbial carbon and rate of change. Looking at the two maps side by side makes it difficult to visualize both components simultaneously for the same grid locations. While we understand the view of the reviewer, we see additional information being communicated through that figure and would like to keep it in the manuscript.

We redesigned Figure 3 to address the suggested minor edits.

Supp. Figure 2: here showed the results based on the model predictions, how about the results from the raw data (i.e., 762 samples shown in Figure 1)?

Response: We added Supp. Figure 4 in relation to the comment below to illustrate the model behavior along a range of predicted values. Raw data points were also added to that figure.

Could you please show the partial dependence plots for each of the 10 predictor variables? An example of the partial dependence plots please see the reference below:

Warner, D. L., Benjamin Bond-Lamberty, Jinshi Jian, Emma Stell, and Rodrigo Vargas. "Spatial predictions and associated uncertainty of annual soil respiration at the global scale." *Global Biogeochemical Cycles* 33, no. 12 (2019): 1733-1745.

Response: We agree that this is useful to complement Supp. Figure 3 and added partial dependence plots as Supp. Figure 4.

REVIEWERS' COMMENTS

Reviewer #1 (Remarks to the Author):

The authors have satisfactorily dealt with all my concerns, including detailed responses and new analyses. The manuscript has improved considerably and I have no further comments.

Reviewer #2 (Remarks to the Author):

Please see the attachment for my comments.

I am still not convinced that soil microbial carbon is decreasing under global warming. However, I have to admit that the authors have done a good job in revising this paper and most of the issues raised by the reviewers have been well addressed. I still have few comments but mainly minor, please see below.

Line 301-302: provide units for RMSE, overall R^2 needs to be explained.

Figure 5: in the previous comment, I was referring to figure 5, please see the figure below to understand what I want to say.

a The position of this figure label is far from the figure

b has a frame line
but a does not have one

Why panel b is much smaller than a? can both panels present in same size for better visualization?

Supplemental Figure 3: add unit for Nitrogen content, soil organic carbon, and precipitation in the x-axis label.

Supplemental Figure 4: add unit for Nitrogen content, soil organic carbon, and precipitation in the x-axis label. For Clay content, x-axis shows a range of 0-500? Should divide by 10? Check the same issue for soil pH, sand content, and mean temperature.

REVIEWERS' COMMENTS

Reviewer #1 (Remarks to the Author):

The authors have satisfactorily dealt with all my concerns, including detailed responses and new analyses. The manuscript has improved considerably and I have no further comments.

Response: We thank Reviewer #1 for their constructive comments in the previous revision that improved the manuscript. We are pleased that we were able to address their concerns satisfactorily.

Reviewer #2 (Remarks to the Author):

Please see the attachment for my comments. [The attachment content was transcribed below.]

I am still not convinced that soil microbial carbon is decreasing under global warming. However, I have to admit that the authors have done a good job in revising this paper and most of the issues raised by the reviewers have been well addressed. I still have few comments but mainly minor, please see below.

Response: We agree with that the relationship between soil microbial carbon and climate is complex, and that further research on the topic will help to elucidate these interactions. We thank the reviewer for their helpful comments and tried to address all the remaining comments.

Line 301-302: provide units for RMSE, overall R² needs to be explained.

Response: We added units and additional information to clarify the text: "The random forest model used for temporal predictions was validated by comparing the observed and predicted values of microbial carbon concentrations. The root-mean-square error (RMSE) was 65.0 mmol kg⁻¹, and the cross-validated R² for out-of-bag predictions was 0.40, while the overall R² was 0.90. The observed microbial carbon values correlated to the fitted values, with a Pearson's r value of 0.59 (Figure 2; p < 0.001)." Lines 305-308.

Figure 5: in the previous comment, I was referring to figure 5, please see the figure below to understand what I want to say. The position of [panel a] is far from the figure. b has a frame line but a does not have one. Why is panel b so much smaller than a? can both panels present in same size for better visualization? [See original PDF from reviewer for figure.]

Response: Sorry for the confusion with figure numbers. We now separated the two panels of Figure 5 into Figures 5 and 6, as the needed sizes were different.

Supplemental Figure 3: add unit for Nitrogen content, soil organic carbon, and precipitation in the x-axis label.

Response: Thank you for noticing these omissions. We added units for all variables in Supplementary Figure 3.

Supplemental Figure 4: add unit for Nitrogen content, soil organic carbon, and precipitation in the x-axis label. For Clay content, x-axis shows a range of 0-500? Should divide by 10? Check the same issue for soil pH, sand content, and mean temperature.

Response: We thank the reviewer for their comment. There was indeed an issue with displaying the scale of these variables. We fixed the issue and added units.